# *Plasmodium* DEH is ER-localized and crucial for oocyst mitotic division during malaria transmission

David S Guttery[1,2,*] , Rajan Pandey[1,3,*] , David JP Ferguson[4,5] , Richard J Wall[1,6] , Declan Brady[1] , Dinesh Gupta[3] , Anthony A Holder[7] , Rita Tewari[1]

Cells use fatty acids (FAs) for membrane biosynthesis, energy storage, and the generation of signaling molecules. 3-hydroxyacyl-CoA dehydratase—DEH—is a key component of very long chain fatty acid synthesis. Here, we further characterized in-depth the location and function of DEH, applying in silico analysis, live cell imaging, reverse genetics, and ultrastructure analysis using the mouse malaria model *Plasmodium berghei*. DEH is evolutionarily conserved across eukaryotes, with a single DEH in *Plasmodium* spp. and up to three orthologs in the other eukaryotes studied. DEH-GFP live-cell imaging showed strong GFP fluorescence throughout the life-cycle, with areas of localized expression in the cytoplasm and a circular ring pattern around the nucleus that colocalized with ER markers. Δ*deh* mutants showed a small but significant reduction in oocyst size compared with WT controls from day 10 postinfection onwards, and endomitotic cell division and sporogony were completely ablated, blocking parasite transmission from mosquito to vertebrate host. Ultrastructure analysis confirmed degeneration of Δ*deh* oocysts, and a complete lack of sporozoite budding. Overall, DEH is evolutionarily conserved, localizes to the ER, and plays a crucial role in sporogony.

## Introduction

Malaria remains one of the world's deadliest infectious diseases. Caused by apicomplexan parasites belonging to genus *Plasmodium*, malaria is responsible for great socioeconomic loss to affected countries. According to World Health Organisation reports, there were 212 million clinical cases of malaria infection and 429,000 deaths in 2015 (WHO, 2018), and growing resistance against existing drugs has further intensified this problem. Hence, there is a growing need to identify new biological pathways and proteins essential for parasitic growth and development in both human and non-human hosts, which could act as suitable drug targets. *Plasmodium* parasites have a complex life cycle and require two hosts to complete the life cycle: vertebrates (during asexual stages) and invertebrates (during sexual stages) (Aly et al, 2009). The disease is transmitted to vertebrate hosts by infected female *Anopheles* mosquitoes, which inject sporozoites into the dermis of the vertebrate host during a blood meal. The parasite enters the circulation, and once it invades liver cells, and subsequently erythrocytes, undergoes several rounds of atypical closed mitotic cell division through multiple rounds of DNA replication and asynchronous nuclear division (termed schizogony) to produce merozoites that invade erythrocytes. During this period of cyclic asexual proliferation in the blood stream, a subpopulation undergoes gametocytogenesis to develop into male and female gametocytes, which are transmitted to a mosquito during its blood meal. Gamete development, fertilization, and zygote formation occur in the mosquito midgut, leading to the differentiation of an infective ookinete, which undergoes meiosis, penetrates the midgut wall, and develops into an oocyst on the basal surface of the midgut, where further rounds of closed mitotic cell division occur. Thousands of sporozoites develop within each oocyst, and then egress into the haemocoel to invade the salivary glands and begin a new life cycle.

Lipid metabolism includes essential cellular processes that use fatty acids (FAs) in membrane biosynthesis, energy storage, and the generation of signaling molecules. FA synthesis is a four-step cyclic process that results in the addition of two carbons to the chain with each cycle. In humans, the process involves condensation of acyl-CoA with malonyl-CoA to produce 3-ketoacyl-CoA (catalyzed by one of seven FA elongases), reduction of 3-ketoacyl-CoA by a 3-ketoacyl-CoA reductase (KAR) to 3-hydroxyacyl-CoA, dehydration of 3-hydroxyacyl-CoA to 2,3-trans-enoyl-CoA (catalyzed by one of four 3-hydroxyacyl-CoA dehydratase isoenzymes; HACD1-4 also known

---

[1]School of Life Sciences, Queens Medical Centre, University of Nottingham, Nottingham, UK   [2]The Leicester Cancer Research Centre, College of Life Sciences, University of Leicester, Leicester, UK   [3]Translational Bioinformatics Group, International Centre for Genetic Engineering and Biotechnology, Aruna Asaf Ali Marg, New Delhi, India   [4]Department of Biological and Medical Sciences, Faculty of Health and Life Science, Oxford Brookes University, Oxford, UK   [5]Nuffield Department of Clinical Laboratory Science, University of Oxford, John Radcliffe Hospital, Oxford, UK   [6]Wellcome Trust Centre for Anti-Infectives Research, School of Life Sciences, University of Dundee, Dundee, UK   [7]The Francis Crick Institute, London, UK

Correspondence: dsg6@le.ac.uk; rita.tewari@nottingham.ac.uk
*David S Guttery and Rajan Pandey contributed equally to this work

as DEH in some systems [see below]), and finally reduction to an acyl-CoA with two additional carbon chain units by 2,3-trans-enoyl-CoA reductase (TER) (Kihara, 2012). HACD1-4 was initially annotated as PTPLA, PTPLB, PTPLAD1, and PTPLAD2, respectively, because of their similarities to the yeast *PHS1* gene product (Ikeda et al, 2008). HACD1 and HACD2 genes restored growth of yeast SAY32 *PHS1*–defective cells, indicating that they are functional homologues of Phs1p, that is, 3-hydroxyacyl-CoA dehydratases. Furthermore, studies have indicated that HACD1 has an essential role in myoblast proliferation and differentiation (Lin et al, 2012), with HACD1-deficient cell lines displaying S-phase arrest, compromised G2/M transition, and retarded cell growth. Studies of the *Arabidopsis thaliana* Phs1p homologue PASTICCINO2 or PAS2 showed the protein has an essential role in very long chain fatty acid (VLCFA) synthesis (Bach et al, 2008), as well as being essential during cell division, proliferation, and differentiation (Bellec et al, 2002). Furthermore, *Arabidopsis* PAS2 complements Phs1p function in a yeast mutant defective for FA elongation (Morineau et al, 2016). PAS2 interacted with FA elongase subunits in the ER and in its absence 3-hydroxyacyl-CoA accumulates, as expected from loss of a dehydratase involved in FA elongation. Similarly, in the yeast *Saccharomyces cerevisiae* VLCFA synthesis is also catalyzed in the ER by a multi-protein elongase complex, following a similar reaction pathway as mitochondrial or cytosolic FA synthesis (Tehlivets et al, 2007).

In apicomplexans, the process of fatty acid synthesis (FAS) and assembly into more complex molecules is critical for their growth and development, while also determining their ability to colonize the host and to cause disease. They acquire lipids through de novo synthesis and through scavenging from the host (Mazumdar & Striepen, 2007), and simple components such as mosquito-derived lipids determine within-host *Plasmodium* virulence by shaping sporogony and metabolic activity, affecting the quantity and quality of sporozoites, respectively (Costa et al, 2018). FAS occurs in the apicoplast via the type II FAS (FASII) pathway, followed by fatty acid elongation (FAE) on the cytoplasmic face of the ER through the elongase (ELO) pathway (Ramakrishnan et al, 2012, 2013) (Fig S1). Studies on whether FAS is essential suggest that different *Plasmodium* spp. have different requirements for these enzymes. In *Plasmodium yoelii*, the FASII enzymes are only essential during liver stages (Yu et al, 2008; Vaughan et al, 2009), whereas in *Plasmodium falciparum*, genetic disruption of the FASII enzymes FabI and FabB/F results in complete abolition of sporogony (van Schaijk et al, 2014). Specifically, day 17–day 23 after mosquito feeding, FabB/F mutant oocysts appeared to degenerate, and protein expressed from the *dhfr* resistance marker fused with *gfp* in PfΔ*fabB*/F deletion mutants was barely detectable using fluorescence microscopy. The enzymatic steps of the ELO process are similar to those in the FASII pathway in the apicoplast (Tarun et al, 2009); however, the growing chain is held by CoA instead of acyl carrier protein. In *Toxoplasma gondii*, the ELO pathway consists of three additional enzymes involved in condensation: ELO-A, ELO-B and ELO-C (Ramakrishnan et al, 2012), with ELO-A and ELO-B engaged in the elongation of de novo–synthesized unsaturated FAs and ELO-C primarily acting on host-derived saturated FAs. In *T. gondii*, the activity of the ELO-pathway is considered an alternative route to FASII-independent ¹⁴C-acetate incorporation (Bisanz et al, 2006) and is engaged in conventional elongation rather than de

novo synthesis (Mazumdar & Striepen, 2007). Indeed, *P. falciparum* parasites with no functional FASII pathway can still elongate FAs, possibly because of the activity of the ELO pathway (Yu et al, 2008). Genetic deletion studies of the ELO enzymes in *Toxoplasma* have suggested functional redundancy (Ramakrishnan et al, 2012), whereas in *Plasmodium*, ELO-A has a crucial role during liver-stage development (Stanway et al, 2019).

In a genome-wide study of *Plasmodium berghei* (Pb) protein phosphatases, we identified 30 phosphatase genes together with one for a predicted protein tyrosine phosphatase-like protein, PbPTPLA, which was shown to be essential for sporozoite formation and completion of the parasite life cycle, but not fully characterized (Guttery et al, 2014). However, despite the original annotation as an inactive PTP-like protein (Andreeva & Kutuzov, 2008; Wilkes & Doerig, 2008; Guttery et al, 2014; Pandey et al, 2014), more recent functional studies indicate that it is a key component of the VLCFA elongation cycle—more specifically the ELO pathway as a 3-hydroxyacyl-CoA dehydratase (DEH) (Stanway et al, 2019). Therefore, to investigate further the role of DEH in *Plasmodium* development, we performed an in-depth genotypic and phenotypic analysis of the protein, using in silico, genetic manipulation, and cell biological techniques. We show evolutionary conservation of DEH in the model organisms examined here. Furthermore, we show that PbDEH is located at the ER and is essential for cell division and parasite budding within oocysts, with its deletion blocking parasite transmission.

# Results

## Phylogenetic analysis reveals that DEH is highly conserved among eukaryotes

Genome-wide analysis showed DEH is present in all the eukaryotic organisms studied here, which includes apicomplexans, fungi, plants, nematodes, insects, birds, and mammals. The number of encoded DEH proteins was shown to vary from one to three in the studied organisms, with *Plasmodium* spp genomes coding for a single DEH. Both *A. thaliana* and *Oryza sativa* encode three DEHs (PAS2 and 2 HACD isozymes) each, as compared with two (HACD1 and HACD2) in *Homo sapiens* (plus two sharing relatively weak similarity—HACD3 and HACD4) and two in *Mus musculus*. Phylogenetic analysis using the neighbor joining method clustered organisms based on their evolutionary relatedness (Fig S2 and Table S1). In addition, the phylogenetic analysis suggests that gene duplication in non-Chordata, Chordata, and plants where there are multiple DEH copies may have happened independently from a single *deh* gene to perform specific functions after divergence during evolution, based on the grouping of all DEH isoforms in the same cluster.

## *Plasmodium* DEH does not contain the canonical PTPLA CXXGXXP motif and is predicted in silico to interact with factors associated with FAE

*P. berghei* (PBANKA_1346500) and *P. falciparum* (Pf; PF3D7_1331600) DEH genes are annotated as PTPLA (pfam04387) (Andreeva &

Kutuzov, 2008; Wilkes & Doerig, 2008; Guttery et al, 2014; Pandey et al, 2014), the criterion for PTPLA being the presence of a PTP active site motif (CXXGXXR) but with arginine replaced by proline (CXXGXXP). However, CLUSTALW alignment of *Pb* and *Pf* protein sequences with the human and mouse HACD1 and HACD2 shows this motif is absent (Fig S3), indicating that *Plasmodium* DEHs cannot be classified as PTP-like proteins. Secondary structure prediction of high confidence showed the presence of six hydrophobic helices followed by coils and the absence of β sheets (Fig S4A). In the absence of a PbDEH crystal structure or a significant template for homology-based 3D modeling revealed by BLASTP, we used I-TASSER (an ab initio threading based tool) to predict the 3D structure of PbDEH. This analysis provided a prediction consistent with the secondary structure and of a three dimensional structural model with six membrane-spanning helices (Fig S4B).

STRING database analysis predicted that PfDEH interacts with FAE and FAS proteins, and many other proteins with an ER location (Fig S5). These proteins include 3-oxo-5-α-steroid 4-dehydrogenase (PBANKA_09127; PF3D7_1135900), stearoyl-CoA δ-9 desaturase (PBANKA_1110700; PF3D7_\0511200), putative long-chain polyunsaturated FAE enzyme (ELO-B, PBANKA_0104700; PF3D7_060590—involved in the FAE pathway), and β-hydroxyacyl-(acyl-carrier-protein) dehydratase (FabZ), involved in stage 3 of FAS in the FASII pathway (Stanway et al, 2019). In addition, interactome analysis revealed DEH interaction with a putative ER membrane protein, Acetyl-CoA transporter protein (PF3D7_1036800).

### DEH is expressed throughout the *Plasmodium* life-cycle stages and localized to the ER

To determine the expression profile and location of PbDEH, we used a single homologous recombination strategy to tag the 3′ end of the endogenous *deh* locus with sequence coding for GFP (Guttery et al, 2014), and then analyzed blood and mosquito stages of the life-cycle for GFP. Strong GFP fluorescence was observed throughout all life-cycle stages analyzed, with areas of localized expression in the cytoplasm and a circular ring formation around the nucleus (Fig 1). Predotar analysis (Small et al, 2004) predicted an ER localization for both PbDEH and PfDEH. Colocalization with ER tracker Red confirmed the DEH-GFP location at the ER, in all parasite stages analyzed (Fig 2A), with subcellular fractionation of blood stage parasites confirming its integral membrane location (Fig 2B).

### PbDEH is essential for mitotic cell division of *Plasmodium* during oocyst development

Previous comparison of Δ*deh* and WT parasite lines highlighted the nonessential role of this gene for blood stage development (Guttery et al, 2014). In this study, we confirmed it is not essential during asexual blood stages, or for zygote development (Fig 3A). However, whereas the overall number of oocysts observed in Δ*deh* and WT lines was not significantly different (Guttery et al, 2014), there was a significant reduction in Δ*deh* GFP-expressing oocysts beginning at day 7 and continuing through day 21 postinfection (Fig 3B and Table S2), with many appearing to be degenerating. Analysis of oocyst size revealed a small (but statistically significant) decrease from day 10 onwards in Δ*deh* lines compared with WT (Fig 3C and Table S2), and by day 21, the vast majority of Δ*deh* oocysts expressed GFP no

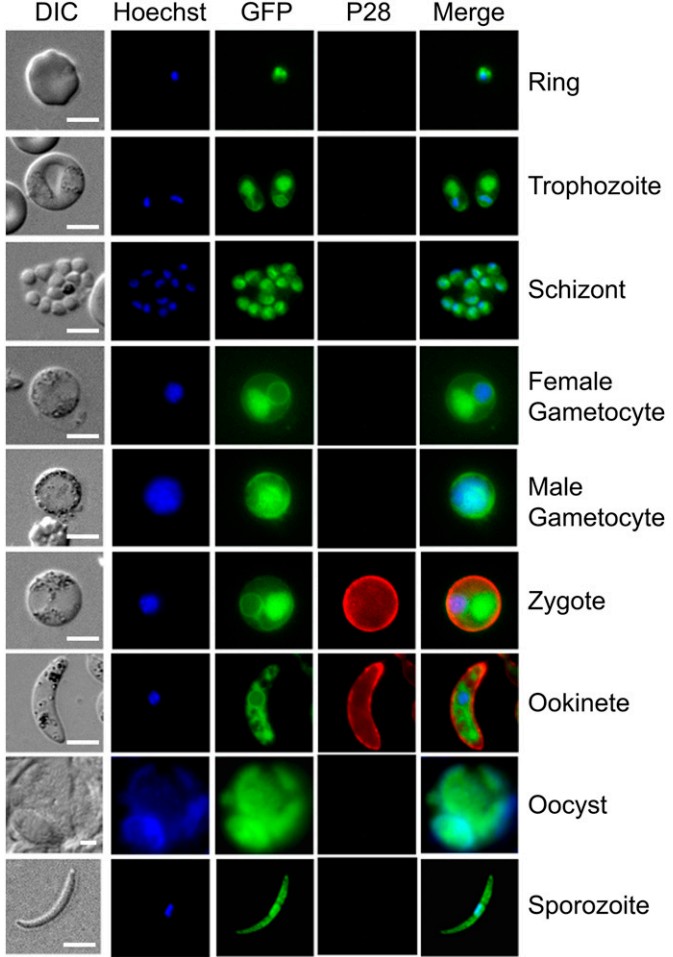

**Figure 1. DEH-GFP protein expression in stages of the parasite life cycle.**
Expression of DEH-GFP in rings, trophozoites, schizonts, gametocytes, zygotes, ookinetes, oocysts, and sporozoites. P28, a cy3-conjugated antibody which recognizes P28 on the surface of zygotes, and ookinetes was used as a marker of the sexual stages. Note that the female gametocyte in this figure has not been activated and is not expressing P28. Scale bar = 5 μm.

longer, and in the few that did GFP was present at very low levels or in fragmented patterns. However, it is important to note that Δ*deh* oocysts that continued to express GFP and showed faint DAPI staining of DNA, were similar in size to WT oocysts; whereas the vast majority of oocysts that reduced in size did not express GFP or stain with DAPI, suggesting they were dead. Analysis of salivary glands from mosquitoes infected with Δ*deh* parasites revealed no sporozoites, in contrast with WT parasite–infected mosquitoes (Fig 3D). Representative examples of Δ*deh* oocyst morphology and lack of sporozoite development at all stages postinfection are shown in Figs 3E and S6, highlighting oocyst degeneration, fragmented GFP expression, and failure to form sporozoites.

### Ultrastructure analysis confirmed oocyst degeneration and apoptotic-like nuclear chromatin condensation in Δ*deh* lines

To investigate further the marked differences in oocyst morphology and complete lack of sporozoite formation, we used electron

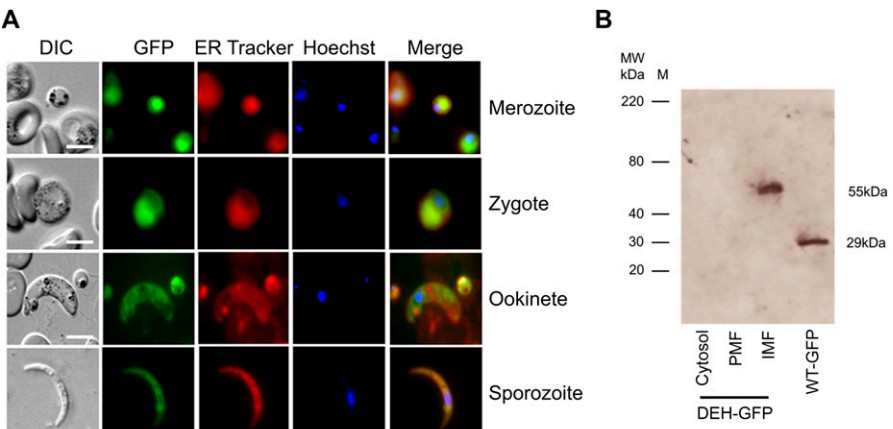

**Figure 2. Co-localization of DEH-GFP and ER tracker.**
**(A)** Analysis of DEH-GFP localization using ER tracker Red in merozoites, zygotes, ookinetes, and sporozoites. Scale bar = 5 µm. **(B)** Anti-GFP Western blot for subcellular localization of DEH-GFP. For WT-GFP, cytosolic GFP is shown. IMF, integral membrane protein fraction; PMF, peripheral membrane protein fraction.

microscopy to compare Δ*deh* and WT lines at 10-, 14-, and 21-d postinfection. At 10 d, the oocysts of Δ*deh* and WT parasites were of a similar size. However, Δ*deh* oocysts showed numerous cytoplasmic vacuoles (Fig 4A) with evidence of dilatation of the nuclear membranes (Fig 4B). At day 14 postinfection, the majority of Δ*deh* oocysts exhibited mid (30%) or advanced (70%) stages of degeneration with increased cytoplasmic vacuolation, dilatated nuclear membranes, and evidence of mitochondrial abnormalities (Fig 4C and D). There was little evidence of retraction of the plasmalemma from the oocyst wall and no evidence that sporozoite inner membrane complex formation had been initiated in any of the 20 oocysts examined. At day 21 postinfection, all Δ*deh* oocysts were in an advanced stage of degeneration—almost completely vacuolated with a few nuclei appearing to have undergone apoptotic-like nuclear chromatin condensation (Fig 4E and F). In contrast, at day 10 postinfection, WT oocysts were completely filled with cytoplasm with numerous nuclear and mitochondrial profiles (Fig 4G and H). At 14 d postinfection, the WT parasites showed a mixture of early (15%), mid (60%), and late (25%) stage oocysts with sporozoites at various stages of development (Fig 4I and J). At day 21 postinfection, the majority of WT oocysts (85%) were late stage with numerous fully formed and free sporozoites (Fig 4K and L), although a few mid-stage (10%) and rare (<5%) degenerated oocysts were observed.

## Discussion

In this study, we examined the location and function of *Plasmodium* DEH using in silico, genetic manipulation and cell biological techniques. Lipid metabolism is essential for cellular function, and includes critical pathways for FA synthesis and elongation. DEH is a 3-hydroxyacyl-CoA dehydratase involved in VLCFA synthesis, which interacts with several elongase units, is located at the ER (Beaudoin et al, 2009; Morineau et al, 2016) and has an essential role during development, differentiation, and maintenance of a number of tissue types (Li et al, 2000; Bellec et al, 2002; Pele et al, 2005).

In our previous protein phosphatome study, a putative, catalytically inactive, PTP-like protein with an essential role during

sporogony was identified (Guttery et al, 2014), which had been classified as a putative PTPLA by others (Andreeva & Kutuzov, 2008; Wilkes & Doerig, 2008; Pandey et al, 2014) based on high sequence similarity and e-score values. However, a recent genome-wide functional screen in *P. berghei* showed that PbPTPLA has an essential role in lipid metabolism, specifically during the ELO pathway as a 3-hydroxyacyl dehydratase (DEH) (Stanway et al, 2019). The specific criterion for a PTP-like protein is the presence of a CXXGXXP motif (i.e., the CXXGXXR motif of PTPs, but with the arginine replaced by proline). However, we show here that this motif is not present in either *P. falciparum* or *P. berghei* protein and this, along with its proven function in lipid metabolism (Stanway et al, 2019), suggests that the classification as a phosphatase-like protein is erroneous. Our in silico interactome analysis suggests that PfDEH interacts with a number of proteins involved in lipid metabolism, confirming previous functional findings (Stanway et al, 2019) and adding further weight to its annotation as a 3-hydroxyacyl-CoA dehydratase.

Studies in mammalian systems have suggested that the ER-bound DEH catalyzes the third of four reactions in the long-chain FA elongation cycle (Ikeda et al, 2008). Our detailed GFP-based localization analyses showed that the protein is expressed strongly throughout all life-cycle stages analyzed, with protein expression at localized areas in the cytoplasm and as a circular ring-like structure around the nucleus. In silico analysis using Predotar and microscopy-based co-localization using ER tracker confirmed the ER location, consistent with a previous study suggesting a role in FAS in *Plasmodium* (Stanway et al, 2019). Phenotypic analysis of DEH function throughout the life cycle confirmed the results of our previous study (Guttery et al, 2014), highlighting that it is essential for oocyst maturation and sporozoite development, but dispensable for asexual blood stage development (Bushell et al, 2017). Time-course analysis at days 7, 14, and 21 after mosquito infection showed that although early-stage Δ*deh* oocysts were comparable in size to WT oocysts, they begin to degenerate at an early stage of development, with a significant decrease in GFP-expressing oocysts even at day 7 postinfection, and as seen previously in other FAE-critical mutant parasites (Stanway et al, 2019). Ultrastructure analysis confirmed that at 14 d postinfection,

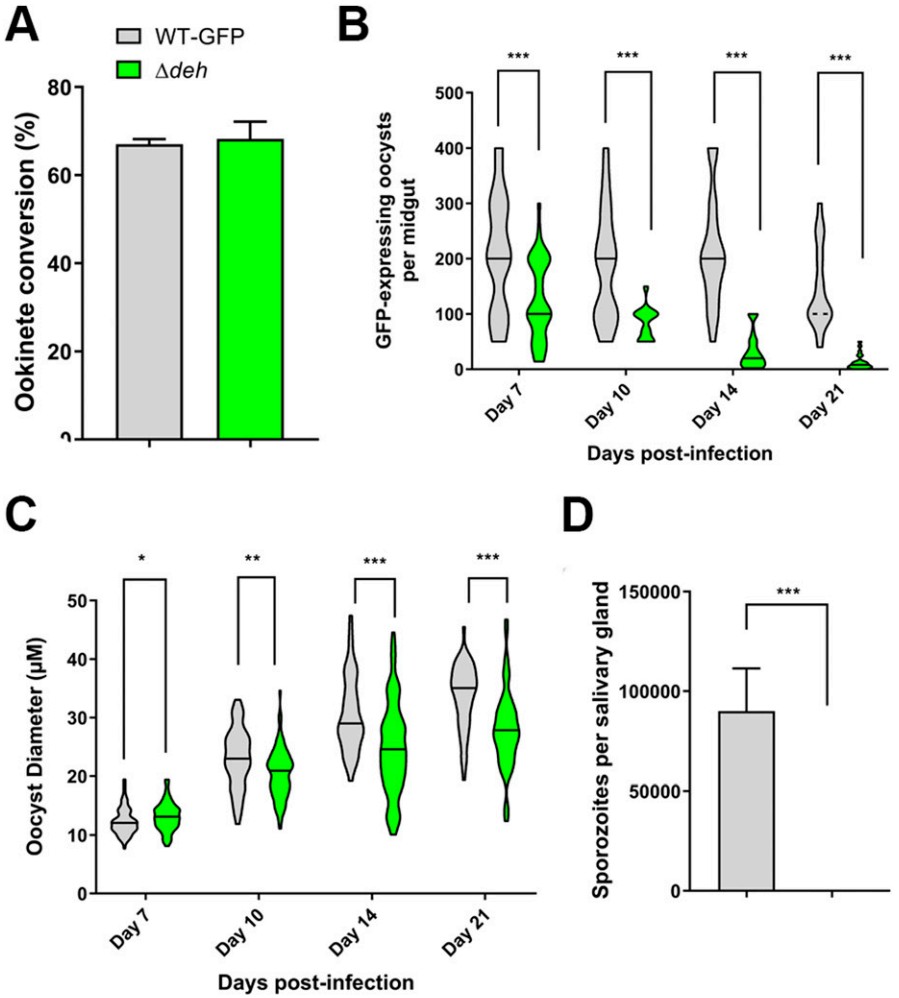

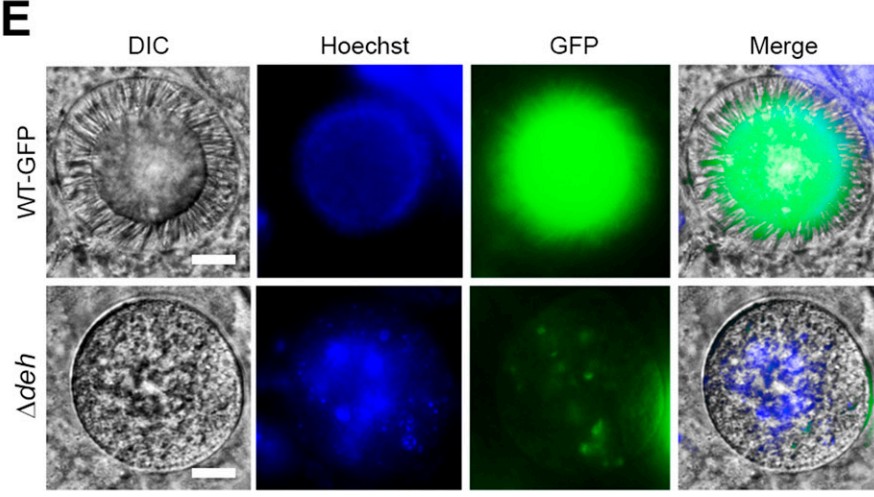

**Figure 3. Phenotypic analysis of Δ*deh* lines.**
**(A)** Ookinete conversion as a percentage in Δ*deh* and WT lines. Ookinetes were identified using the marker P28 and defined as those cells that successfully differentiated into elongated "banana shaped" ookinetes. Bar is the mean ± SEM. n = 3 independent experiments. **(B)** Total number of GFP-positive oocysts per infected mosquito, including normal and smaller oocysts, at 7, 10, 14, and 21 d postinfection for Δ*deh* and WT parasite lines. Bar is the mean ± SEM. n = 3 independent experiments (20 mosquitoes for each). *P* < 0.001 for all time points. **(C)** Individual Δ*deh* and WT oocyst diameters (*μm*) at 7, 10, 14, and 21 d postinfection. Horizontal line indicates the mean from three independent experiments (20 mosquitoes for each) of Δ*deh* and WT. *P < 0.05, **P < 0.01, ***P < 0.001. **(D)** Total number of sporozoites per mosquito from 21 d postinfection salivary glands for Δ*deh* and WT lines. Three independent experiments, n = 20 mosquitoes for each replicate. ***P < 0.001. **(E)** Representative examples of Δ*deh* and WT oocysts (63× magnification) at 21 dpi showing fragmented GFP and Hoechst staining. Scale bar = 20 *μm*.

Δ*deh* oocysts were at an advanced state of degeneration, with no evidence of sporozoite development. Retraction of the oocyst plasmalemma (the parasite plasma membrane) from the oocyst capsule is a crucial first stage in sporozoite development, where sporoblast formation is followed by thousands of sporozoites budding off into the space delineated by the capsule (Aly et al, 2009). A model of this process has been detailed in Burda et al (2017). Our study suggests that initiation of mitosis, which results in

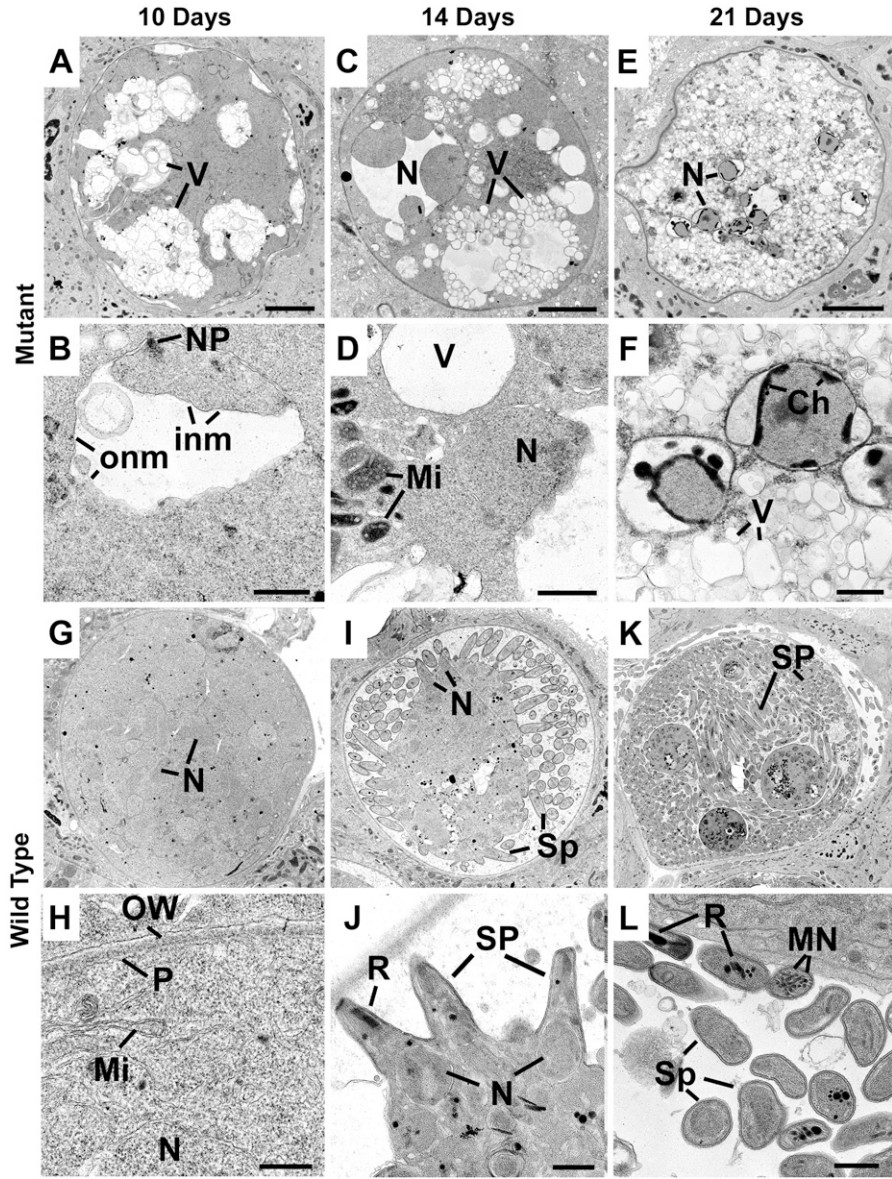

**Figure 4. Ultrastructure analysis of oocyst development in Δ*deh* lines.**
**(A, B, C, D, E, F, G, H, I, J, K, L)** Electron micrographs of Δ*deh* (A, B, C, D, E, F) and WT (G, H, I, J, K, L) parasites at 10 d (A, B, G, H), 14 d (C, D, I, J), and 21 d (E, F, K, L) postinfection. **(A, B, C, D, E, F, G, H, I, J, K, L)** Bars represent 10 μm (A, C, E, G, I, K) and 1 μm (B, D, F, H, J, L). **(A)** Low-power image of an early oocyst showing lucent area made up numerous vacuoles (V). **(A, B)** Detail from a similar stage to that in (A) showing part of the cytoplasm containing a nucleus with a nuclear pole (NP). Note the lucent area due to the separation of the inner (inm) and outer (onm) nuclear membranes. **(C)** Low power image of a mid-stage oocyst showing nuclear swelling (N) and increased numbers of lucent cytoplasmic vacuoles (V). **(D)** Detail part of the cytoplasm showing a swollen nucleus (N), membrane bound lucent vacuoles (V), and mitochondria (Mi) with vesicles embedded in electron dense material. **(E)** Low power image of a late stage oocyst with abnormal nuclei (N) and the cytoplasm filled with electron lucent vacuoles. **(E, F)** Detail from the central region of (E) showing the peripheral location of electron dense chromatin (Ch) typical of apoptotic changes, whereas the cytoplasm consists of numerous vacuoles (V). **(G)** Low power image of an early oocyst (end of growth phase) in which the cytoplasm completely fills the oocyst and contains many nuclear profiles (N). **(H)** Detail of the peripheral cytoplasm limited by the plasmalemma (P) containing mitochondria (M) and nuclei (N). **(I)** Mid-stage oocyst showing the surface formation of numerous sporozoites (Sp). N, nucleus. **(J)** Detail showing partially formed sporozoites (Sp) budding from the surface of the cytoplasmic mass. N, nucleus; R, rhoptry. **(K)** Mature oocysts containing large number of fully form sporozoites (Sp). **(L)** Detail of cross sections through mature sporozoites (Sp) containing rhoptries (R) and micronemes (MN). OW, oocyst wall.

sporozoite development, does not even commence in Δ*deh* oocysts because retraction of the plasmalemma and initiation of daughter inner membrane complex formation is ablated. The phenotype is similar to that of a cyclin-3 mutant (Roques et al, 2015), with defects leading to abnormalities in membrane formation, vacuolation, and subsequent cell death during the later stages of sporogony. However, in contrast to the cyclin-3 mutant, sporogony was completely ablated in Δ*deh* parasites, and no transmission was observed in bite-back experiments. This suggests that the parasite cannot scavenge VLCFA from its mosquito host environment, and that DEH (and therefore the ELO pathway) is critical for oocyst mitotic maturation and differentiation. The cells were unable to progress further to form additional lobes and start sporozoite budding in its absence, although the oocyst size was not grossly affected, suggesting that two independent processes drive oocyst formation and sporogony, respectively.

Although FASII activity is exclusively in the apicoplast (Shears et al, 2015), our study showed that DEH-GFP is located at the ER, suggesting that it is an active component of the ELO pathway. The genes involved in the ELO pathway include members of the ELO family (1, 2, and 3), of which *ELO2* and *ELO3* are involved in keto- and enoyl reduction (Kohlwein et al, 2001). In yeast, the dehydratase step is carried out by the DEH-like homologue, Phs1p, which has also been characterized as a cell cycle protein with mutants defective in the G2/M phase (Yu et al, 2006). Gene KO studies for any ELO proteins are few, with a single genome-wide functional analysis showing that the *P. berghei* homologue of PF3D7_0605900 (a putative long chain polyunsaturated FAE enzyme) is dispensable during the asexual blood stages (Bushell et al, 2017). In addition, a comprehensive analysis of FAE in *Plasmodium* (Stanway et al, 2019) showed that mutants of a ketoacyl-CoA reductase (KCR) have an identical phenotype to our DEH gene KO lines, with normal development of ookinetes and oocysts gradually disappearing over the course of

development, resulting in the complete ablation of sporogony. In contrast, ELO-A (stage 1 of VLCFA synthesis) mutants were critical for liver stage development. This suggests that reduction of ketoacyl-CoA to hydroxyacyl-CoA and subsequent dehydration of hydroxyacyl-CoA to enoyl-CoA (i.e., stages 2 and 3 of VLCFA synthesis) are the most crucial stages for oocyst maturation and sporogony, whereas the lack of a phenotype during sporogony of the ELO-A deletion may suggest functional redundancy and/or a compensatory mechanism such as an overlapping specificity for condensation of malonyl-CoA by either ELO-B or ELO-C, as suggested in *Trypanosoma brucei* (Lee et al, 2006) and *T. gondii* (Ramakrishnan et al, 2012).

In conclusion, our PbDEH analysis using various in silico, in vitro, and in vivo approaches provides important insights into the crucial role DEH plays during VLCFA synthesis, and how disruption of the gene can affect parasite development in the mosquito. Future studies will elucidate further how lipid metabolism in *Plasmodium* can be explored as a viable target for therapeutic intervention.

# Materials and Methods

### Ethics statement

All animal works were performed following ethical approval and was carried out under United Kingdom Home Office Project Licence 40/3344, in accordance with the UK "Animals (Scientific Procedures) Act 1986" and in compliance with "European Directive 86/609/EEC" for the protection of animals used for experimental purposes. 6- to 8-wk-old female Tuck-Ordinary (Harlan) or Swiss Webster (Charles River) outbred mice were used for all experiments.

### Identification of conserved domains and evolutionary lineage

The deduced amino acid sequence of PBANKA_134650 (PbPTPLA), now classified as DEH in the article, was retrieved from PlasmoDB (release 27) (Aurrecoechea et al, 2009). Conserved domains in PbPTPLA (DEH) were identified using the Conserved Domain Database (Marchler-Bauer et al, 2011), the Simple Modular Architecture Research Tool (Schultz et al, 1998) and Protein Family Database (PFAM) (Finn et al, 2008). The deduced amino acid sequence and individual conserved domains were used as BLAST (BLASTP) queries to identify orthologs in PlasmoDB and National Center for Biotechnology Information (NCBI) protein databases. OrthoMCL database (version 5) was used to identify and retrieve *P. berghei* orthologs (Table S1) (Li et al, 2003). Multiple sequence alignment was performed for the retrieved sequences using ClustalW (Larkin et al, 2007). ClustalW alignment parameters included gap opening penalty 10 and gap extension penalty 0.1 for the pairwise sequence alignment; gap opening penalty 10 and gap extension penalty 0.2 was used for multiple sequence alignment. A gap separation distance cutoff of 4 and Gonnet protein weight matrix was used for the alignments. Residue-specific penalty and hydrophobic penalties were used, whereas end gap separation and negative matrix were excluded in the ClustalW alignments. The phylogenetic tree was inferred using the neighbor-joining method, computing the evolutionary distances using the Jones–Taylor–Thornton model for amino acid substitution with the Molecular Evolutionary Genetics Analysis software (MEGA 6.0) (Tamura et al, 2013). Gaps and missing data were treated using a partial deletion method with 95% site coverage cutoff and 1,000 bootstrap replicates to generate a phylogenetic tree. iTOL was used to visualize the phylogenetic tree (Letunic & Bork, 2019). For structure analyses, the secondary structure of the PbDEH was evaluated using PSIPRED (Buchan et al, 2013). I-TASSER, an Iterative threading assembly-based tool, was used to generate 3D structure of PbDEH (Zhang, 2008). The STRING database was used to identify PTPLA interacting proteins (Szklarczyk et al, 2019). For STRING search, we used a cutoff of 0.70 for the parameters of neighborhood, gene fusion, co-occurrence, co-expression, experiments, databases, and text mining results. Predotar (Small et al, 2004) was used for inferring Pf and PbDEH subcellular localization.

### Generation of transgenic parasites and genotype analysis

Details of GFP-tagged PTPLA (termed DEH-GFP in this study) and *deh* (PBANKA_1346500) KO parasite lines (Δ*deh* in this study) are given in Guttery et al (2014). For this study, the KO construct was transfected into the GFPCON wild-type line (Janse et al, 2006), with three clones produced by serial dilution.

### Parasite development in the mosquito

*Anopheles stephensi* mosquitoes (3–6 d old) were allowed to feed on anaesthetized mice infected with either wild type or mutant parasites at comparable gametocytemia as assessed by blood smears. Mosquitoes were dissected post-blood meal, on the days indicated. For midgut and salivary gland sporozoites, organs from 10 to 20 mosquitoes were pooled and homogenized, and released sporozoites were counted using a haemocytometer. For oocyst counts, midguts taken at day 7, 14, and 21 postinfection were harvested, mounted on a slide, and oocysts counted using phase or fluorescence microscopy. To quantify sporozoites per oocyst (the ratio of number of sporozoites to number of oocysts), an equal number of mosquitoes from the same cage were used to count the number of oocysts and sporozoites. This number varied among experiments but at least 20 mosquitoes were used for each count. For light microscopy analysis of developing oocysts, at least 20 midguts were dissected from mosquitoes on the indicated days and mounted under Vaseline-rimmed cover slips. ER tracker Red (Thermo Fisher Scientific, Cat. no. E34250) was used to perform co-localization studies according to manufacturer's instructions. Images were collected with an AxioCam ICc1 digital camera fitted to a Zeiss AxioImager M2 microscope using a 63x oil immersion objective. Statistical analyses were performed using GraphPad prism software.

### Electron microscopy

The guts from mosquitoes harvested at 10, 14, and 21 d postinfection were dissected and fixed in 2.5% glutaraldehyde in 0.1 M phosphate buffer and processed for electron microscopy (Guttery et al, 2012). For quantitation, between 10 and 20 oocysts were examined for each group.

### Subcellular fractionation of parasite lysates and detection of DEH

Immunoprecipitation and subcellular fractionation of lysates containing GFP-tagged DEH was performed as described previously (Guttery et al, 2014). WT-GFP was used as the control protein in all experiments. In summary, cells from mouse blood infected with the DEH-GFP–expressing parasite were pelleted and then lysed in hypotonic buffer (10 mM Tris–HCl, pH 8.4, 5 mM EDTA) containing protease inhibitors (Roche, Cat. no. 04693159001), freeze/thawed twice, incubated for 1 h at 4°C, and then centrifuged at 100,000$g$ for 30 min. The supernatant was collected as the soluble protein fraction (cytosol). The pellet was resuspended and washed in carbonate solution (0.1M $Na_2CO_3$, pH 11.0) containing protease inhibitors (Roche, Cat. no. 04693159001), and after incubation for 30 min at 4°C, the sample was centrifuged again at 100,000$g$ for 30 min. The supernatant was saved as the peripheral membrane protein fraction (PMF). The residual pellet was solubilized in 4% SDS and 0.5% Triton X-100 in PBS, to form the integral membrane protein fraction (IMF). Samples from these three fractions, containing equal amounts of protein, were then analyzed by Western blot using anti-GFP polyclonal rabbit antibody (Invitrogen, Cat. no. A11122) and the Western Breeze Chemiluminescence Anti-Rabbit kit (Invitrogen, Cat. no. WB7106).

## Supplementary Information

## Acknowledgements

We thank Julie Rodgers for maintaining the insectary breeding colony and Ms Jessica Stock for assistance in Western blot analysis. Funding: This project was funded by a Medical Research Council (MRC) Investigator Award and MRC project grants to R Tewari (G0900109, G0900278, and MR/K011782/1). R Pandey was supported by Newton Bhabha PhD Placement Program (BT/IN/UK/DBT-BC/2015-16). AA Holder is supported by the Francis Crick Institute (FC10097) which receives its core funding from Cancer Research UK (FC10097), the UK Medical Research Council (FC10097), and the Wellcome Trust (FC10097). The funders had no role in study design, data collection and analysis, decision to publish, or preparation of the manuscript.

### Author Contributions

DS Guttery: conceptualization, formal analysis, investigation, methodology, and writing—original draft.
R Pandey: data curation, formal analysis, investigation, methodology, and writing—original draft.
DJP Ferguson: data curation, formal analysis, investigation, methodology, and writing—original draft.
RJ Wall: data curation, formal analysis, investigation, methodology, and writing—original draft.
D Brady: formal analysis, investigation, and methodology.
D Gupta: formal analysis, investigation, and writing—original draft.
AA Holder: conceptualization, data curation, formal analysis, supervision, funding acquisition, investigation, methodology, and writing—original draft.
R Tewari: conceptualization, resources, data curation, formal analysis, supervision, funding acquisition, investigation, methodology, writing—original draft, and project administration.

### Conflict of Interest Statement

The authors declare that they have no conflict of interest.

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
