## [Reviewer comments · Life Science Alliance]

Life Science Alliance

Plasmodium DEH is ER-localized and crucial for oocyst mitotic division during malaria transmission

David Guttery, Rajan Pandey, David Ferguson, Richard Wall, Declan Brady, Dinesh Gupta, Tony Holder, and Rita Tewari

DOI: <https://doi.org/10.26508/lsa.202000879>

Corresponding author(s): David Guttery, UNIVERSITY OF LEICESTER

Review Timeline:

Submission Date:	2020-08-17
Editorial Decision:	2020-09-25
Revision Received:	2020-10-13
Editorial Decision:	2020-10-15
Revision Received:	2020-10-15
Accepted:	2020-10-15

Scientific Editor: Shachi Bhatt

Transaction Report:

September 25, 2020

Re: Life Science Alliance manuscript #LSA-2020-00879

Dr. David GUTTERY
UNIVERSITY OF LEICESTER
Leicester Royal Infirmary
Leicester LE2 7LX
United Kingdom

Dear Dr. Guttery,

Thank you for submitting your manuscript entitled "Plasmodium DEH is crucial for oocyst mitotic division but not cell size during malaria transmission" to Life Science Alliance. The manuscript was assessed by expert reviewers, whose comments are appended to this letter. As you can see, the reviewers are generally enthusiastic about the work, but do express some concerns about the advance and request additional data to support some of the claims listed in the manuscript. We would like to invite you to submit a revised manuscript to Life Science Alliance that further clarifies the advance provided by this study, and addresses the rest of the reviewers' concerns.

We would be happy to discuss the individual revision points further with you should this be helpful. While you are revising your manuscript, please also attend to the below editorial points to help expedite the publication of your manuscript. Please direct any editorial questions to the journal office.

The typical timeframe for revisions is three months. Please note that papers are generally considered through only one revision cycle, so strong support from the referees on the revised version is needed for acceptance. When submitting the revision, please include a letter addressing the reviewers' comments point by point.

Thank you for considering Life Science Alliance as an appropriate venue for your research. We look forward to receiving your revised manuscript.

Sincerely,

Shachi Bhatt, Ph.D.
Executive Editor
Life Science Alliance

- A letter addressing the reviewers' comments point by point.
- An editable version of the final text (.DOC or .DOCX) is needed for copyediting (no PDFs).
- High-resolution figure, supplementary figure and video files uploaded as individual files: See our detailed guidelines for preparing your production-ready images, <https://www.life-science-alliance.org/authors>
- Summary blurb (enter in submission system): A short text summarizing in a single sentence the study (max. 200 characters including spaces). This text is used in conjunction with the titles of papers, hence should be informative and complementary to the title and running title. It should describe the context and significance of the findings for a general readership; it should be written in the present tense and refer to the work in the third person. Author names should not be mentioned.

B. MANUSCRIPT ORGANIZATION AND FORMATTING:

Reviewer #1 (Comments to the Authors (Required)):

This study by Guttery and colleagues shows that the *Plasmodium berghei* protein PBANKA_1346500 (PTPLA, DEH) localizes in the ER and plays a vital role in oocyst differentiation to sporozoites. This protein was wrongly annotated as a protein phosphatase. The same authors have already reported a critical role for this protein in sporozoite development in a previous paper (Guttery et al., 2014), while another study already reported the protein to be involved in fatty acid biosynthesis (Stanway et al., 2019). The new study brings these two previous pieces of information together and expands it with some nice subcellular localisation and a more detailed phenotyping of the knockout mutant. However, there is insufficient advancement of our knowledge of this molecule for publication in LSA. The study is suited for a parasitology journal.

specific points:

The world malaria report 2018 seems to be written by a Dr W. H. Organisation. Please correct.

The introduction would benefit from a figure outlining the enzymatic pathways for FAS and ELO in apicomplexa and other organisms, and show where DEH fits in.

The paper would also benefit from a cartoon showing the predicted DEH structure and perhaps indicating where the catalytic sites lie if known, to complement the multiple alignment provided.

EM data needs to be quantified. How many oocysts were examined from how many different mosquitoes and what proportion shows the phenotypes reported.

Reviewer #2 (Comments to the Authors (Required)):

1. This paper details an assessment of Plasmodium Berghe DEH temporal and spatial expression patterns in the blood and mosquito stages of the life cycle and the phenotypic analysis of DEH knockout on the life cycle. Brief analysis of the knockout had previously been published and this body of work follow up on that analysis. The advance to the field is ultimately the fact that DEH is essential for P. berghei sporozoite production.
2. The authors claim that delta DEH oocysts are similar in size to WT but their actual data does not corroborate this claim. This needs to be addressed.
3. With regard to penmanship, text changes, data presentation and statistics. I have extensively edited the PDF I was given for the review using the freely available editing tools provided by Adobe and have sent this PDF to the journal and asked the journal to share this document with the authors.

Reviewer #1 (Comments to the Authors (Required)):

This study by Guttery and colleagues shows that the *Plasmodium berghei* protein PBANKA_1346500 (PTPLA, DEH) localizes in the ER and plays a vital role in oocyst differentiation to sporozoites. This protein was wrongly annotated as a protein phosphatase. The same authors have already reported a critical role for this protein in sporozoite development in a previous paper (Guttery et al., 2014), while another study already reported the protein to be involved in fatty acid biosynthesis (Stanway et al., 2019). The new study brings these two previous pieces of information together and expands it with some nice subcellular localisation and a more detailed phenotyping of the knockout mutant. However, there is insufficient advancement of our knowledge of this molecule for publication in LSA. The study is suited for a parasitology journal.

We thank the reviewer for these comments; however, we feel that we have advanced knowledge in this area due to our illustration of degenerating oocysts through loss of GFP expression by day 21 post-infection, an analysis that we did not perform in our original screen (Guttery et al. 2014). Further, through ultrastructure analyses we have provided the first evidence of oocyst degeneration with increased cytoplasmic vacuolation, dilated nuclear membranes, and evidence of mitochondrial abnormalities, extending our phenotypic analysis. We have also shown that PbDEH localizes to the ER in all stages analysed, which again is the first evidence of this in Plasmodium. We feel that these data are of interest to the general readership of LSA, since they expand on results obtained in other Apicomplexa.

specific points:

The world malaria report 2018 seems to be written by a Dr W. H. Organisation. Please correct.

We have corrected this error

The introduction would benefit from a figure outlining the enzymatic pathways for FAS and ELO in apicomplexa and other organisms, and show where DEH fits in.

We thank the reviewer for this suggestion and have included this in the supplementary figures (Figure S1).

The paper would also benefit from a cartoon showing the predicted DEH structure and perhaps indicating where the catalytic sites lie if known, to complement the multiple alignment provided.

We thank the reviewer for this suggestion and have included this in the supplementary figures (Figure S4) as well as highlighting this in the results section. Unfortunately, we were unable to identify the catalytic site.

EM data needs to be quantified. How many oocysts were examined from how many different mosquitoes and what proportion shows the phenotypes reported.

We have now quantified the EM data and included them in the results and methods sections.

Reviewer #2 (Comments to the Authors (Required)):

1. This paper details an assessment of *Plasmodium Berghei* DEH temporal and spatial

expression patterns in the blood and mosquito stages of the life cycle and the phenotypic analysis of DEH knockout on the life cycle. Brief analysis of the knockout had previously been published and this body of work follow up on that analysis. The advance to the field is ultimately the fact that DEH is essential for *P. berghei* sporozoite production.

2. The authors claim that delta DEH oocysts are similar in size to WT but their actual data does not corroborate this claim. This needs to be addressed.

We thank the reviewer for this suggestion and have subsequently altered the text to reflect it. It now reads "Analysis of oocyst size revealed a small (but statistically significant) decrease from day 10 onwards in Δ deh lines compared to WT (Figure 3C), and by day 21 the vast majority of Δ deh oocysts expressed GFP no longer, and in the few that did GFP was present at very low levels or in fragmented patterns."

3. With regard to penmanship, text changes, data presentation and statistics. I have extensively edited the PDF I was given for the review using the freely available editing tools provided by Adobe and have sent this PDF to the journal and asked the journal to share this document with the authors.

We thank the reviewer for these suggestions and have altered the manuscript based on their recommendations and comments.

Points in PDF:

1. in your introduction, you refer to these (DEH) enzymes as HACD which seems like a perfectly reasonable nomenclature and yet you refer to the plasmodium product as DEH. i don't understand why.

The HACD nomenclature is indeed used to describe the human orthologues; however, in many Apicomplexan species (including Plasmodium – Stanway, 2019) it is termed DEH. Therefore, we wished to continue this nomenclature.

2. aren't there four in humans?

There are 2 highly conserved HACD enzymes in humans (HACD1 and 2) and 2 sharing relatively weak similarity – HACD3 and HACD4.

3. i don't believe this statement is accurate, they are smaller at every time point where measurements took place according to your own data.

We have addressed this in the manuscript as described above.

4. it is not clear from reading on, what the two stages are.

We have altered the paragraph to read: "Lipid metabolism includes essential cellular processes that use fatty acids (FAs) in membrane biosynthesis, energy storage and the generation of signaling molecules. FA synthesis is a four-step cyclic process that results in the addition of two carbons to the chain with each cycle."

5. there is no evidence provided in the manuscript sited that VLCFA is crucial for sporogony - only that fatty acid sythesis is required. studies were not carried out to determine if there was a decrease in VLCFA in the mutant parasites.

We thank the reviewer for highlighting this and have removed this sentence.

6. i think here you should also briefly mention the other three enzymes involved in the elongation process that you mention above - are they present in the apicomplexan genomes, have any been characterized. this would round off the intro well.

We thank the reviewer for this suggestion and have briefly discussed these enzymes in the introduction (page 6).

7. you did not analyze expression throughout the life cycle - you have no liver stage data

We thank the reviewer for highlighting this and have adjusted all instances to say "in all stages analysed".

8. your day 14 delta DEH image looks rather similar to wildtype and thus it is somewhat misleading when one compared Fig 3B with 3C

We thank the reviewer for highlighting this. These images have been removed to avoid confusion.

9. your decrease in oocyst size, albeit small, is statistically significant and thus i don't fully understand why you seem to dismiss this finding. since you don't show us the number of oocysts that don't express GFP for either WT or the delta DEH, this figure is hard to interpret. you claim that the vast majority did not but show no tabulated data. one could assume that all the oocysts are dead as they do not form sporozoites? also, it is an unfair comparison, in my mind, to compare the size of only GFP positive oocysts for the knockout with GFP positive oocysts for the wild type. if you were unable to score the oocysts in this way, how would you ever have made these distinctions. ultimately, the knockout produces smaller oocysts, if you take all the oocysts into account no? However, you claim otherwise in your abstract.

We thank the reviewer for this suggestion and have included the tabulated data in Table S2. We have also included the small but significant decrease in oocyst size in the abstract, as well as highlighting this in the results section.

10. your labeling of the panels in figure 4 is confusing and you should refer to each panel in order in the text. you jump from 4b to 4g and then back to 4c etc.

We thank the reviewer for highlighting this and have re-written this section to improve the flow.

11. Dilitated - is this correct?

Thank you for highlighting this. This was a typographical error and now reads a "dilatated".

October 15, 2020

RE: Life Science Alliance Manuscript #LSA-2020-00879R

Dr. David GUTTERY
UNIVERSITY OF LEICESTER
Leicester Royal Infirmary
Leicester LE2 7LX
United Kingdom

Dear Dr. GUTTERY,

Thank you for submitting your revised manuscript entitled "Plasmodium DEH is ER-localized and crucial for oocyst mitotic division during malaria transmission". We would be happy to publish your paper in Life Science Alliance pending final revisions necessary to meet our formatting guidelines.

Along with the points listed below, please also attend to the following:

- please use the [10 author names, et al.] format in your references (i.e. limit the author names to the first 10)
- please rename the 'Methods' section as 'Materials and Methods'
- please provide more information on the materials used in the experimental procedures - e.g. please clarify the catalog numbers of protease inhibitors or anti-GFP ab used in sub-cellular fractionation experiment, etc.
- For consistency in labeling all the figures, please edit the Fig 4 panel labels to be 4A, 4B... instead of 4a, 4b... please also edit the callouts in the manuscript text accordingly
- please remove the description for Fig 3F from the Fig 3 legend

A. FINAL FILES:

B. MANUSCRIPT ORGANIZATION AND FORMATTING:

Sincerely,

Shachi Bhatt, Ph.D.
Executive Editor
Life Science Alliance
<https://www.life-science-alliance.org/>
Tweet @SciBhatt @LSAJournal

October 15, 2020

RE: Life Science Alliance Manuscript #LSA-2020-00879RR

Dr. David GUTTERY
UNIVERSITY OF LEICESTER
Leicester Royal Infirmary
Leicester LE2 7LX
United Kingdom

Dear Dr. GUTTERY,

Thank you for submitting your Research Article entitled "Plasmodium DEH is ER-localized and crucial for oocyst mitotic division during malaria transmission". It is a pleasure to let you know that your manuscript is now accepted for publication in Life Science Alliance.

DISTRIBUTION OF MATERIALS:

I hope you found the review process to be constructive and are pleased with how the manuscript was handled editorially. We look forward to future exciting submissions from your lab.

Sincerely,

Shachi Bhatt, Ph.D.
Executive Editor
Life Science Alliance
<https://www.life-science-alliance.org/>
Tweet @SciBhatt @LSAjournal